# SMRT Sequencing Technology Was Used to Construct the *Batocera horsfieldi* (Hope) Transcriptome and Reveal Its Features

**DOI:** 10.3390/insects14070625

**Published:** 2023-07-11

**Authors:** Xinju Wei, Danping Xu, Zhiqian Liu, Quanwei Liu, Zhihang Zhuo

**Affiliations:** College of Life Science, China West Normal University, Nanchong 637002, China; weixinjuxx@foxmail.com (X.W.); xudanping@cwnu.edu.cn (D.X.); qnhtvxhp319123@foxmail.com (Z.L.); quanwei66977@163.com (Q.L.)

**Keywords:** *Batocera horsfieldi*, full-length transcripts, functional annotation, Illumina RNA-seq, single-molecule real-time (SMRT) sequencing

## Abstract

**Simple Summary:**

*Batocera horsfieldi* (Hope) is an important wood-boring pest in China. This pest primarily infests tree trunks by feeding on the woody tissue, creating a network of interconnected tunnels within the trunk. These tunnels become blocked with insect feces and wood debris, causing damage, decay, and even the death of the host plant’s tissues. In this study, single-molecule real-time sequencing (SMRT) and Illumina RNA-seq technologies were employed to conduct full-length transcriptome sequencing of male and female adults of *B. horsfieldi*. A total of 20,356,793 subreads (38.26 G, clean reads) were generated, including 432,091 circular consensus sequences and 395,851 full-length non-chimera reads. Clustering and redundancy removal of the full-length non-chimera reads resulted in 39,912 consensus reads. Additionally, functional annotation was performed on a total of 84,650 transcripts in seven different databases. This study provides an important foundation for future exploration of gene regulation in the interaction between *B. horsfieldi* and host plants using RNA interference (RNAi), and it offers a scientific basis for the prevention and control of *B. horsfieldi*.

**Abstract:**

*Batocera horsfieldi* (Hope) (Coleoptera: Cerambycidae) is an important forest pest in China that mainly infests timber and economic forests. This pest primarily causes plant tissue to necrotize, rot, and eventually die by feeding on the woody parts of tree trunks. To gain a deeper understanding of the genetic mechanism of *B. horsfieldi*, this study employed single-molecule real-time sequencing (SMRT) and Illumina RNA-seq technologies to conduct full-length transcriptome sequencing of the insect. Total RNA extracted from male and female adults was mixed and subjected to SMRT sequencing, generating a complete transcriptome. Transcriptome analysis, prediction of long non-coding RNA (lncRNA), coding sequences (CDs), analysis of simple sequence repeats (SSR), prediction of transcription factors, and functional annotation of transcripts were performed in this study. The collective 20,356,793 subreads (38.26 G, clean reads) were generated, including 432,091 circular consensus sequences and 395,851 full-length non-chimera reads. The full-length non-chimera reads (FLNC) were clustered and redundancies were removed, resulting in 39,912 consensus reads. SSR and ANGEL software v3.0 were used for predicting SSR and CDs. In addition, four tools were used for annotating 6058 lncRNAs, identifying 636 transcription factors. Furthermore, a total of 84,650 transcripts were functionally annotated in seven different databases. This is the first time that the full-length transcriptome of *B. horsfieldi* has been obtained using SMRT sequencing. This provides an important foundation for investigating the gene regulation underlying the interaction between *B. horsfieldi* and its host plants through gene editing in the future and provides a scientific basis for the prevention and control of *B. horsfieldi*.

## 1. Introduction

*Batocera horsfieldi* (Hope) is an important forest pest in China that attacks trees by boring into the stems. It belongs to the family Cerambycidae of the order Coleoptera and is also known as *Batocera lineolata* [1]. The pest is mainly distributed abroad in Vietnam, Japan, India, Myanmar, and other countries [2], while domestically it is distributed in Hebei, Beijing, Zhejiang, Fujian, Jiangxi, Hubei, Hunan, Taiwan, Guangdong, Guizhou, Sichuan, Yunnan, and other places [3]. After hatching, the clouded leopard moth larvae begin to feed on the cambium of the host plant, then enter the xylem and continuously bore into the wood, causing damage, rot, and even the death of the host plant tissue [4]. Therefore, it greatly affects the growth and development of the host plant, leading to a decline in fruit quality [5]. Its adults appear in early summer and feed on the branches of the host plant until they reach sexual maturity [6]. *B. horsfieldi* has a wide variety of host species, including *Populus tomentosa*, *Juglans regia*, *Eucalyptus robusta*, *Fraxinus chinensis*, and *Castanea mollissima*, among others [7,8]. *B. horsfieldi* is an insect with a fairly complete protection mechanism. Its strong adaptability and high reproductive capacity cause serious harm to plant growth and the ecological environment. Due to the concealment of its larvae, it is difficult to control, and traditional chemical control methods are rarely effective [6,9]. Currently, research on *B. horsfieldi* mainly focuses on predicting its habitat distribution [3], investigating its impact on hosts [10,11,12], and analyzing the antennal transcriptome and olfactory-related genes of adult insects [13]. These studies have laid the foundation for exploring new control strategies for *B. horsfieldi*.

Transcriptome data can reflect information on cell responses, gene functions, evolution, and other biological processes, revealing different biological processes at the molecular level [14,15,16]. Transcriptome analysis is a high-throughput technology that can simultaneously determine the expression levels of a large number of genes in an organism, thereby revealing the complex mechanisms of insect gene expression and regulation. Through transcriptome analysis, we can gain a deeper understanding of the genetic characteristics and growth and development patterns of insects, providing strong evidence and references for future insect research. In addition, transcriptome analysis can help us discover new genes, reveal metabolic pathways and adaptive mechanisms, and deepen our understanding of insect taxonomy and ecology. Therefore, transcriptome analysis has significant implications for insect research and is expected to further promote the development of basic and applied research on insects, providing new ideas for further research on *B. horsfieldi*. In recent years, transcriptome sequencing has gradually been used in gene expression analysis of the different developmental stages of Coleoptera [17,18], such as *Nicrophorus orbicollis* [19], *Tribolium castaneum* (Herbst) [20], *Callosobruchus chinensis* (L.) [21], and *Lasioderma serricorne* [22], which have all used transcriptome library analysis. Transcriptome library construction has also been widely used in the transcriptome analysis of longhorn beetles, such as *Anoplophora nobilis* and *Monochamus alternatus*, to capture specific genes and conduct functional analysis [13]. In addition, transcriptome and olfactory-related gene analysis of the antennae of *B. horsfieldi* adults have enriched the gene database of Coleoptera and laid the foundation for further research on the behavioral regulation and control of *B. horsfieldi* at the molecular level.

However, there are incomplete splice transcripts in short-read transcriptome sequencing, making it difficult for current short-read sequence prediction programs to accurately predict gene structure [23]. Additionally, using low-quality transcripts obtained from short sequencing may lead to incorrect annotation [24,25]. Second-generation sequencing technology has some limitations, including shorter read lengths and the inability to cover the entire transcriptome, which presents significant challenges to the genome assembly process [26,27]. However, third-generation sequencing technologies represented by PacBio have effectively addressed these issues. Single-molecule sequencing technology, also known as SMRT (single-molecule real-time) sequencing, utilizes its advantage of ultra-long read lengths to directly obtain complete transcripts containing 5′UTR, 3′UTR, and polyA tails without the need for interruption and splicing. This enables accurate analysis of structural information, such as alternative splicing and fusion genes in reference genome species, thus overcoming the challenges of shorter and incomplete transcript splicing in non-reference genome species. In addition, SMRT sequencing technology can also use second-generation sequencing data for transcript-specific expression analysis, thereby obtaining more comprehensive annotation information. As one of the third-generation high-flow sequencing techniques, SMRT has been widely applied in the transcript sequence and analysis of various species, such as *Rhynchophorus ferrugineus* [16], *Hyphantria cunea* (Drury) [28], *Agasicles hygrophila* (Selman and Vogt) [15], *Bactrocera dorsalis* (Hendel) [29], *Odontotermes formosanus* (Shiraki) [30], *Sogatella furcifera* (Horvath) [31], and *Rhopalosiphum padi* (L.) [32]. As far as we know, there are currently few reports of full-length transcriptome sequencing of *B. horsfieldi* for gene expression analysis at different stages of development.

By combining short-read transcriptome sequencing (Illumina RNA-seq) and full-length transcriptome sequencing (PacBio Iso-seq), we have obtained for the first time the complete full-length transcriptome of *B. horsfieldi*. This provides favorable conditions for the comprehensive analysis of the transcriptome information of *B. horsfieldi*. Next, we annotated the complete full-length transcriptome, predicted coding sequences (CDS), analyzed simple sequence repeat sequences, and conducted transcription factor analysis. In addition, we analyzed lncRNAs and other splicing events. It is worth mentioning that we have provided a complete full-length transcriptome for the gene study of *B. horsfieldi*, which provides a reference for further analysis of the gene expression profile of *B. horsfieldi* and a valuable resource for future molecular biology research on *B. horsfieldi*.

## 2. Materials and Methods

### 2.1. Sample Collection and Preparation

The *B. horsfieldi* samples used in this study were collected from China West Normal University, Nanchong City, Sichuan Province. We only selected newly hatched *B. horsfieldi* adults that had not fed or reproduced and were in good physical condition. After collecting the adult insects, we separated the samples into female and male adults according to their gender. The female and male adults were then placed separately in stainless steel rearing cages (60 cm × 60 cm × 60 cm) at a room temperature of 25 ± 2 °C, with a relative humidity of 75–80%, and a light cycle of 12 L:12 D in an artificial climate chamber. They were fed with poplar leaves at room temperature for backup. Then, one male and one female adult insect were selected, and their RNA was extracted and mixed. The mixed overall sample was used for subsequent RNA sequencing, with three biological replicates conducted. All collected samples were frozen in liquid nitrogen and stored at a constant temperature of −80 °C for future experimental use.

### 2.2. Library Preparation and SMRT Sequencing

Total RNA samples were isolated using the RNeasy Plus Mini Kit (Qiagen, Valencia, CA, USA). RNA degradation and contamination were examined using 1% agarose gel electrophoresis. The purity of RNA was determined using the Nanodrop (Nanodrop Products, Rockville, MD, USA) by measuring the OD260/280 ratio. The purity of RNA was also assessed using the RNA IQ assay kit with the Qubit Fluorometer (Life Technologies, Carlsbad, CA, USA). The integrity of RNA was measured using the 2100 Agilent Bioanalyzer system (Agilent Technologies, Santa Clara, CA, USA). The purified RNA products were sent to Beijing Novogene Bioinformatics Technology Co., Ltd., Beijing, China for SMRTbellTM library preparation and sequencing. First, the full-length cDNA required for sequencing was synthesized using the SMARTerTM PCR cDNA Synthesis Kit (TaKaRa USA, Inc., Mountain View, CA, USA). Then, high-quality large-scale library amplification products were purified to 1–6 kb using the BluePippin Size Selection System. The selected full-length cDNA underwent damage repair, end repair, and SMRT dumbbell ligation to form the SMRTbell template library. Polymerase was added to the SMRTbell template library. The Iso-Seq library was prepared according to the Isoform Sequencing Protocol (Iso-Seq) using the Clontech SMARTer PCR cDNA Synthesis Kit and the BluePippin Size Selection System protocol as described by Pacific Biosciences (PN 100-092-800-03). Finally, single-molecule real-time sequencing was performed on the latest third-generation sequencing platform, PacBio RSII.

### 2.3. SMRT Sequencing Data Processing

The sequences were aligned using SMRT software (version 5.0). The circular consensus sequence (CCS) was obtained from subread BAM files (parameters: min_length 200; max_drop_fraction 0.8; no_polish TRUE; min_zscore-9999; min_passes 1; min_predicted_accuracy 0.8; max_length 18,000) and outputted in CCS.BAM file format. The full-length reads contain a 5′ primer, a 3′ primer, and a poly(A) tail. The integrity of transcripts can be assessed based on the presence of a 5′ primer, a 3′ primer, and a poly(A) tail in CCS reads; the CCS reads are clustered into full-length and non-full-length reads. The iterative clustering for error correction (ICE) clustering analysis was performed using FLNC to obtain consensus isoforms. The quiver algorithm [33] (parameters: hq_quiver_min_accuracy 0.99; bin_by_primer false; bin_size_kb 1; qv_trim_5p 100; qv_trim_3p 30) was then used to perform accuracy correction on the consensus isoforms to identify high-quality isoforms. The full-length transcripts were corrected using LoRDEC software v0.8 [34] and Illumina RNA-seq. Finally, the CD-HIT software v4.5.88 was used to remove redundancy and similar sequences from the high-quality transcripts to obtain non-redundant transcripts.

### 2.4. Functional Annotation of Transcripts

To obtain comprehensive functional gene information for the transcriptome of *B. horsfieldi* adult males and females, various databases were used to annotate the non-redundant genes. The BLAST software v.2.2.23 [35] was employed with an E-value threshold set at less than 1^−10^ to compare the obtained transcripts against the NCBI non-redundant protein (Nr) sequence database [36], NCBI non-redundant nucleotide sequence database, Protein Families Database (Pfam) [37], Clusters of Orthologous Groups of Proteins database (KOG) [38], Swiss-Prot [39], Kyoto Encyclopedia of Genes and Genomes (KEGG) [40], and Gene Ontology (GO) [41]. The Pfam database (https://www.ebi.ac.uk/interpro/, accessed on 25 April 2023) was analyzed using Hmmscan. Then, the Blast2GO v2.5 software was used to perform annotation analysis of GO based on the protein annotation results from the Pfam database.

### 2.5. CDS Prediction

The ANGEL pipeline is capable of performing ANGEL [42] long-reads and determining the protein-coding sequences (CDS) of full-length complementary deoxyribonucleic acids (cDNAs). In this study, we tested the ANGEL pipeline using protein sequences from *B. horsfieldi* and its closely related species and then performed CDS prediction analysis on the given sequences. Ultimately, transcripts that include both the 5′- and 3′-UTRs (untranslated regions) and the complete CDSs are defined as full-length transcripts.

### 2.6. SSR Analysis

Transcriptome SSR detection was performed using MISA [43] (version 1.0, default parameters). The minimum repeat times for each unit size were as follows: 1–10, 2–6, 3–5, 4–5, 5–5, and 6–5. For example, for 1–10, at least 10 repeats of a single nucleotide unit were required for detection; for 2–6, at least 6 repeats of a dinucleotide unit were required.

### 2.7. Transcription Factor (TF) Analysis

Plant transcription factors were predicted using iTAK software v1.6 and animal factors were predicted using the animal TFDB 2.0 database [44]. For species that have already been collected in the database, the transcript factor is directly filtered if it is an Ensembl genid, and for genes that are not Ensembl genids, the BLASTX v2.2.31 screening is carried out through the known sequence of transcription factor proteins of the species and the database; for species not collected in the database, the identification is based on the Pfam file of the translation factor family, using hmmsearch (Version V3.3.2).

### 2.8. lncRNAs Analysis

Long non-coding RNAs (lncRNAs), which do not encode proteins, can be screened for coding potential using four tools: CNCI [45], CPC2 [46], Pfam-scan [37], and PLEK [47]. These tools are used to analyze the transcripts and determine whether they have coding potential. Transcripts with coding potential are filtered out, and the intersection of non-coding transcripts identified by the four analysis software programs is taken as the final predicted lncRNA result.

## 3. Results

### 3.1. Transcriptome Analysis Was Performed using Pacbio Sequencing

A total of 462,134 polymerase readings were obtained using PacBio SMRT sequencing technology. After removing the adapter sequences form the polymerase reads, the remaining sequence is called the subreads. A total of 20,356,793 subreads were obtained from the 46.31 Gb of data, with an average length of 2275 bp. A CCS sequence is a consistency sequence obtained from the subreads in each zero-mode waveguide (ZMW) hole through its comparison correction. After removing adapters and artifacts, a total of 432,091 CCS sequences were generated, with 397,316 full-length reads. By classifying CCS, a total of 34,775 non-full-length (nFL) sequences were identified; 1465 were full-length chimera reads, and 395,851 were full-length non-chimera reads (FLNC), with an average length of 2407 bp. The proportion of the FLNC number to the CCS number is 91.61%. Details of the above data are shown in Table 1. Using the hierarchical n*log(n) algorithm, FLNCs were clustered and redundant, with a total of 39,912 consensus reads, an alkaline base number of 97,462,894, and an average length of 2442 bp. N90 is 1652 bp, and N50 is 2631 bp.

In addition, the samples were subjected to replicate sequencing using Illumina Novaseq 6000, generating a total output of 41.16 G raw reads. After filtering, the total clean reads amounted to 38.26 G, as shown in Table 2. The LoRDEC software v0.8 (http://atgc.lirmm.fr/lordec, accessed on 25 April 2023) was used to correct the third-generation PacBio data using the more accurate Illumina reads. After correction, the average length was 2442 bp; N90 was 1652 bp; and N50 was 2630 bp. And, by using the CD-Hit [48] software sequence ratio for clustering and removing redundant and similar sequences, the consensus transcripts were eventually clustered into 15,233 transcript books for subsequent analysis; the comparison of data before and after redundancy removal is shown in Table 3. As shown in Table 4, 67.1% of unigenes have one allelic type, while 32.9% of unigenes have two to ten allelic types. From Figure 1, it can be observed that the length of the genes after redundancy removal is mainly distributed in the range of 1–5 k.

### 3.2. Functional Annotation of B. horsfieldi

To comprehensively understand the gene function information of *B. horsfieldi*, we annotated 15,233 transcripts in seven databases, including Swiss-Prot, KOG, GO, NR, NT, Pfam, and KEGG. Summing up, 14,459, 12,619, 14,016, 10,788, 10,783, 11,202, and 10,783 transcripts were annotated in the NR, Swiss-Prot, KEGG, KOG, GO, NT, and Pfam databases, respectively. In addition, at least 14,791 transcripts were annotated in at least one database, and 7057 transcripts were annotated in all databases (Figure 2).

The Non-Redundant Protein Database (NR) is a comprehensive protein database created and maintained by the NCBI. The annotations in the database include species information, making them useful for species classification. Comparing gene sequences with those of closely related species in the NR database can provide information on the similarity and function of genes in a given species. *B. horsfieldi* was compared to the protein sequences of closely related species in the NR database. According to the results shown in Figure 3, a total of 14,459 transcripts were annotated in the NR database. The top six species with the most annotations were Anoplophora glabripennis (83.74%), *T. castaneum* (3.08%), *Sinocyclocheilus rhinocerous* (1.83%), *Sinocyclocheilus anshuiensis* (1.53%), *Marmota marmota* (1.29%), and *Cyprinus carpio* (1.29%). This provides important evidence for future in-depth understanding of the protein structure and function of *B. horsfieldi*.

KOG is a system for evolutionary relationships based on the complete genome-encoded proteins of bacteria, algae, and eukaryotes. To better analyze the functional aspects of the *B. horsfieldi* transcriptome, this study compared the *B. horsfieldi* transcripts with the KOG database. A total of 10,788 transcripts were successfully annotated (Figure 4), which can be categorized into 26 functional categories. The most annotated functional category was “general function prediction only” with 2149 annotations, accounting for 19.9% of the total annotated transcripts in KOG. The next two categories were “signal transduction mechanisms” with 1782 annotations, representing 16.5%, and “posttranslational modification, protein turnover, chaperones” with 1086 annotations, representing 10%. This provides a basis for analyzing the evolutionary role of this species.

GO stands for Gene Ontology, which is an internationally standardized gene function classification system. By annotating the full-length transcripts of *B. horsfieldi* using the GO database, 10,783 transcripts were successfully classified into three major categories: Cellular Component, Molecular Function, and Biological Process (Figure 5). In the Biological Process category, the cellular process (4897) has the largest proportion, followed by the metabolic process (4588) and the single-organism process (3372). Additionally, we found that some genes were annotated as biological regulation (2037), regulation of biological process (1980), localization (1804), response to stimulus (1381), and signaling (1016) terms. In the Cellular Component category, genes involved in cell (2330), cell part (2330), organelle (1706), membrane (1633), membrane part (1545), and macromolecular complex (1142) were the most abundant. In the Molecular Function (MF) category, binding (7081) and catalytic activity (4565) are the two most abundant subcategories of the annotated transcripts.

The KEGG database is a collection of pathways that systematically analyze the metabolic pathways of genes and compounds in cells, as well as the functions of these gene products. In the case of *B. horsfieldi*, a total of 10,783 transcripts were annotated in the KEGG metabolic pathways. These transcripts can be categorized into six major classes: Cellular Processes, Environmental Information Processing, Genetic Information Processing, Human Diseases, Metabolism, and Organismal Systems, which consist of 44 subcategories. As shown in Figure 6, Human Diseases, Metabolism, and Organismal Systems are the top three categories with the highest proportion. Specifically, there are a total of 4284 genes involved in the Human Diseases-related pathways, among which 510 genes are predicted to be involved in infectious disease: viral, 727 genes are predicted to be involved in cardiovascular disease, and 961 genes are predicted to be involved in cancers: an overview. Second, among the related pathways in organismal systems, the four pathways with the most abundant genes are the nervous system (368 genes), the immune system (617 genes), the endocrine system (820 genes), and the digestive system (314 genes). Moreover, the entire 2472 annotated genes were involved in the Metabolism pathway. The two most enriched pathways were lipid metabolism (337 genes) and carbohydrate metabolism (462 genes). Meanwhile, in terms of Environmental Information Processing, there are a total of 1465 genes involved in signal transduction, 112 genes involved in membrane transport, and 82 genes involved in signaling molecules and interaction, indicating a large number of genes involved in these processes. Likewise, some genes with fewer numbers were annotated in Cellular Processes and Genetic Information Processing.

### 3.3. CDS Predictions

CDS (coding sequence) refers to the sequence that encodes a protein product. In this study, the obtained gene fragments were subjected to coding prediction using the prediction software ANGEL. The predicted CDS results are shown in Figure 7, indicating that the length distribution of CDS ranges from approximately 204 to 5820 nt, with the majority falling between 204 and 4000 nt. As the length of the transcripts increases, the number of transcripts decreases.

### 3.4. Identification of Transcription Factors

Transcription factors (TFs) are a class of proteins that can interact with specific DNA sequences and regulate gene expression [49]. They play a crucial role in various biological processes and are an important component of the transcriptional regulatory system. Using the existing data of *B. horsfieldi*, we predicted a total of 636 transcription factors, among which Zf-C2H2 (203, 31.92%), ZBTB (86, 13.52%), TF_bZIP (42, 6.60%), and bHLH (41, 6.45%) were the top four transcription factor families (Figure 8). These transcription factors lay the foundation for the regulatory mechanism of *B. horsfieldi*.

### 3.5. SSR Analysis

Simple sequence repeats (SSRs), also known as short tandem repeats or microsatellite markers, are a type of repetitive sequence widely distributed in eukaryotic genomes. They are usually composed of a few nucleotides (1–6) in repeat units with a length of several tens of nucleotides. We used MISA software (version 1.0) with default parameters. A total of 5540 SSR loci were identified, among which the most common type was mono-nucleotide motifs (3789, 68.39%), followed by tri-nucleotides (968, 17.47%), di-nucleotide motifs (748, 13.50%), tetra-nucleotides (24, 0.43%), penta-nucleotides (8, 0.18%), and hexa-nucleotide motifs (3, 0.05%) (Figure 9). The identification of SSR loci establishes the foundation for future assessments of genetic diversity in this species.

### 3.6. lncRNA Forecasts

Long-chain noncoding RNA (lncRNA) is a class of RNA with a length greater than 200 nt that does not encode proteins. Due to the limitations of library construction principles, we could only obtain lncRNA with a polyA tail. We used four tools, CNCI [45], PLEK [47], CPC2 [46], and Pfam [37], to identify unique transcripts without protein-coding potential (i.e., lncRNAs). CNCI identified 1606 lncRNAs, PLEK identified 1155 lncRNAs, CPC2 identified 3221 lncRNAs, and Pfam identified 3836 lncRNAs (Figure 10). Meanwhile, we compared the length distribution of lncRNA and mRNA and found that the average length of mRNA was slightly shorter than that of lncRNA (Figure 11). There is a certain correlation between lncRNA and mRNA that can be observed.

## 4. Discussion

Transcriptome sequencing has become one of the primary means for investigating the mechanisms of gene expression regulation, thanks to the continual advancements and enhancements in sequencing technology of recent years. Compared with second-generation sequencing (short-read transcriptome sequencing), third-generation sequencing (full-length transcriptome sequencing) has overcome many challenges through the technology of obtaining full-length transcripts without PCR amplification through assembly [50,51]. Currently, SMRT sequencing technology has been widely applied in multiple fields, including microbial 16S rRNA gene sequencing, microbial genome assembly, transcriptome sequencing, methylation analysis, and genome resequencing [26]. In addition, third-generation sequencing has made significant contributions in multiple fields. It can accurately reflect the transcriptome information of the sequenced species, detect various alternative splicing forms, and discover more splicing sites and alternative splicing events, thereby improving the accuracy of gene function annotation. Moreover, third-generation sequencing can discover new functional genes, supplement existing genome annotations, and promote further research. Furthermore, it can accurately analyze fusion genes, homologous genes, superfamily genes, and allelic genes, providing more possibilities for the study of variant genes and evolution. According a research paper by [52], full-length transcriptome sequencing was performed on *Oxya chinensis*, *Acrida cinerea*, *Atractomorpha sinensis*, *Manis javanica* [53], and *R. ferrugineus* [16] using the PacBio RS II platform. The results showed that the full-length transcriptome obtained higher transcript integrity and better quality compared to the transcriptome obtained through second-generation sequencing and could be used for subsequent transcriptome annotation and analysis. Although PacBio Iso-Seq, as a representative of single-molecule real-time sequencing technology, has the advantages of sequencing during synthesis and fast sequencing speed, the original sequencing data has a relatively high error rate (10–15%) [54,55], which needs to be corrected by second-generation sequencing [56].

One advantage of SMRT sequencing is that it can provide a novel understanding of full-length sequences, gene structure, and gene function. The study on common wheat has shown that SMRT sequencing has great potential for genome annotation and gene function research [57]. In the present research, PacBio Iso-Seq and Illumina RNA-Seq were used to sequence and analyze a mixed RNA sample of male and female *B. horsfieldi* adults. A total of 38.26G clean reads were obtained, including 432,091 CCS, of which 395,851 were identified as FLNC with an average length of 2442 bp, N90 of 1652 bp, and N50 of 2631 bp. After CD-Hit redundancy removal, 15,233 transcripts were obtained for subsequent analysis. In *B. horsfieldi*, the amount of transcriptome data obtained through second-generation sequencing is much higher than in other coleopteran insects at different developmental stages, such as *Hypothenemus hampei* (average length of 1609.92 bp, N50 of 2427 bp) [58] and *M. alternatus* (average length of 819 bp, N50 of 1590 bp) [59]. In addition, 97.09% of the transcripts were successfully functionally annotated, which was significantly higher than other second-generation sequencing coleopterans, such as *Holotrichia parallela* [60] and *Henosepilachna vigintioctopunctata* [61].

In gene annotation research, classifying a large number of new transcripts can help obtain gene function information. Updating and collecting homologous protein sets can be used for genome functional annotation of new sequencing data, including those for complex eukaryotes and whole genome evolution studies [38]. Genome sequencing has revealed that most genes involved in core biological functions are conserved across all eukaryotes [41]. In the SMRT sequencing results of the full-length transcriptome of *B. horsfieldi*, a total of 10,783 full-length transcripts were successfully annotated in the GO database. Among them, most of the transcripts were related to cellular processes, followed by cellular components and molecular functions. In addition, 14,016 transcripts were annotated to 40 KEGG pathways, among which signal transduction, cancers (overview), the endocrine system, and transport and catabolism had the most transcripts. Furthermore, the KOG annotation results showed that the transcripts related to “general function prediction only” and “signal transduction mechanisms” were the most abundant. The gene annotation results indicate that the new transcripts may be related to the above functions.

Some studies have indicated that the process of insect host recognition is influenced by various factors, such as the active components present in the volatile substances emitted by host plants [62,63]. These active components, when mixed in specific proportions, can regulate insect behavior. *B. horsfieldi* shows a distinct antennal response to the volatile compounds emitted by the host plants, *Viburnum awabuki* [10] and *Rosa cymosa* Tratt [12]. Although the genome sequence of *B. horsfieldi* has been published [6,13], its transcriptome features and the structures of mRNA and lncRNA have not been deeply analyzed. In the study of *B. horsfieldi*, the discovery and functional research of lncRNA can provide us with a new perspective to help us understand more about the gene regulation mechanism, especially in the aspect of mutual regulation between non-coding RNA and coding proteins. lncRNA is a class of non-protein-coding transcripts with a length of more than 200 nt that plays an important role in regulating gene expression at various levels. Many studies have shown that non-coding RNAs also play important roles in physiological functions, such as biological growth and development regulation and abiotic stress [64,65]. According to the data in Figure 10, a total of 6058 lncRNAs were predicted, of which 1606, 1155, 3221, and 3836 lncRNA transcripts were predicted by the CNCI, CPC, PLEK, and Pfam, respectively. In addition, non-coding RNAs appear to be more species-specific and may provide more appropriate evidence for the study of biological evolution [66]. In recent years, a large number of lncRNAs have been discovered in insects and animals, such as *Agrilus zanthoxylumi*, *Portunus trituberculatus*, and *Nilaparvata lugens*. Therefore, we have reason to believe that the predicted lncRNAs in this study will help to further reveal the biological characteristics of *B. horsfieldi* [66]. Research on lncRNAs and transcription factors (TFs) can help us better understand the impact of gene regulation on the growth, development, metabolic regulation, and environmental stress of *B. horsfieldi*, which will be helpful for future research and applications in related fields.

## 5. Conclusions

In this study, we used second-generation sequencing (Illumina RNA-seq) to correct the third-generation sequencing (PacBio Iso-Seq) for full-length transcriptome sequencing of *B. horsfieldi* and analyzed the related transcripts. After filtering low-quality sequencing reads, self-correction, and redundancy removal, a total of 15,233 full-length transcripts were successfully generated. We performed gene annotation, CDS prediction, SSR analysis, and TF and lncRNA prediction. These results not only contribute to the improvement of the genome annotation information of *B. horsfieldi* but also provide a valuable foundation for the study of its gene function and for the growth and development of other Coleoptera insects in the future.

## Figures and Tables

**Figure 1 insects-14-00625-f001:**
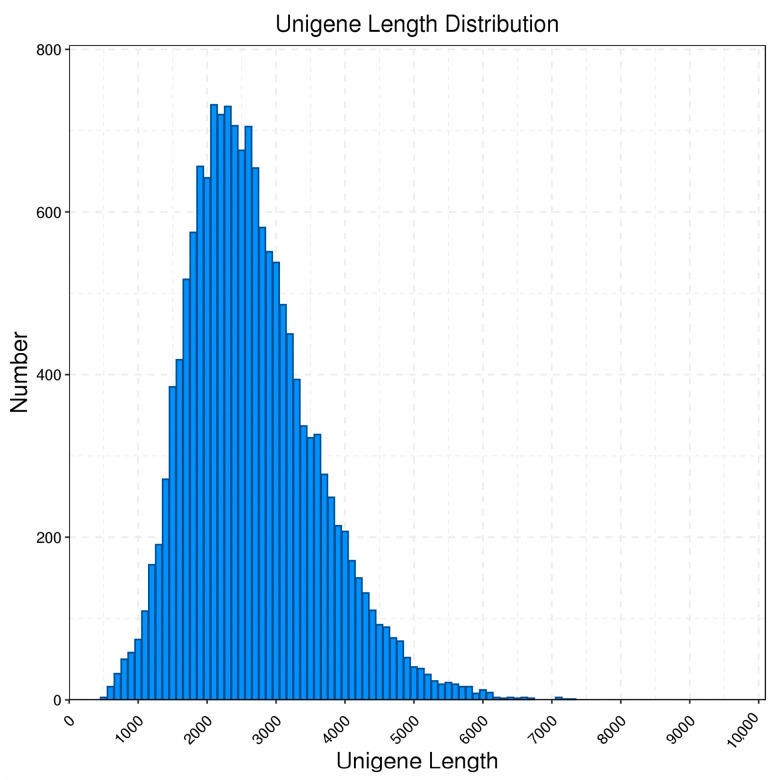
Length distribution of unigenes obtained from PacBio Iso-Seq in *B. horsfieldi*.

**Figure 2 insects-14-00625-f002:**
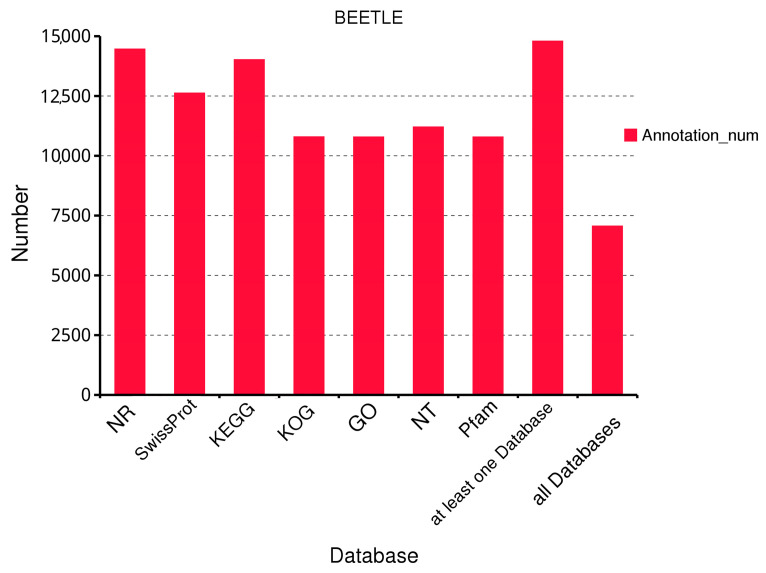
Functional annotation of *B. horsfieldi* transcripts in seven databases. NR is a non-redundant protein database. Swiss-Prot is a manually annotated and reviewed protein sequence database. KEGG stands for Kyoto Encyclopedia of Genes and Genomes. KOG is a eukaryotic orthologous gene database. GO is the Gene Ontology. NT is the NCBI’s non-redundant nucleotide sequence database. Pfam is a protein family database.

**Figure 3 insects-14-00625-f003:**
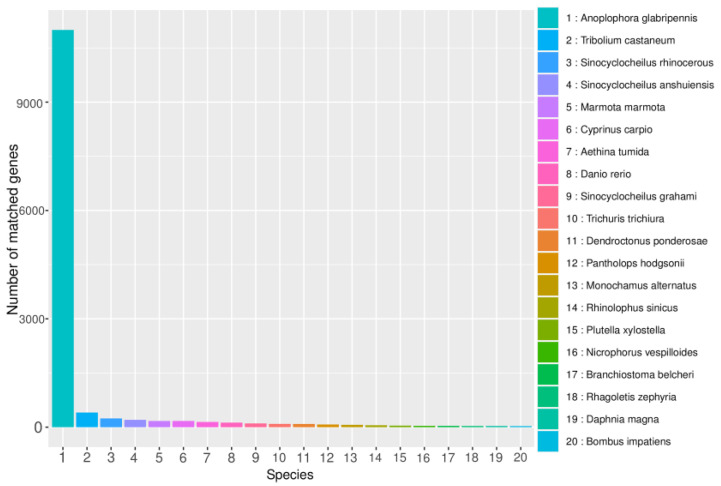
NR database annotation of *B. horsfieldi* genes. The X-axis represents the categories of closely related species, and the Y-axis represents the number of transcripts.

**Figure 4 insects-14-00625-f004:**
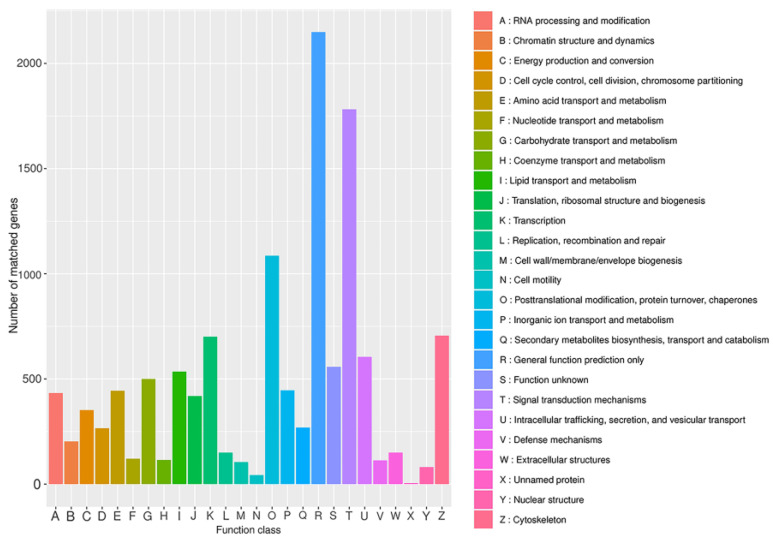
KOG annotation of *B. horsfieldi* transcripts. The x-axis represents the KOG category, and the y-axis represents the number of transcripts.

**Figure 5 insects-14-00625-f005:**
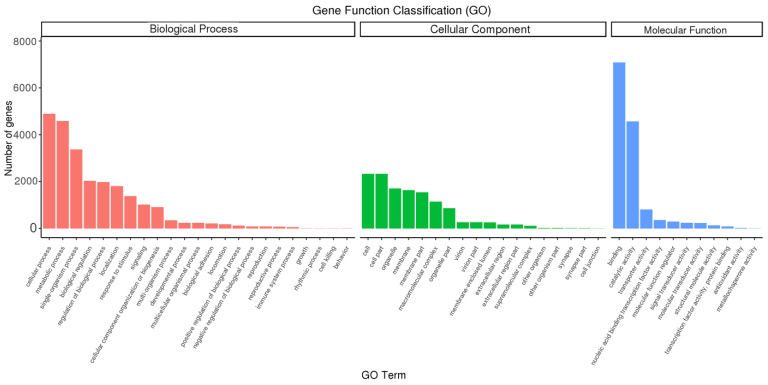
Gene Ontology (GO) annotation of *B. horsfieldi* genes. The x-axis represents the GO category, and the y-axis represents the number of transcripts.

**Figure 6 insects-14-00625-f006:**
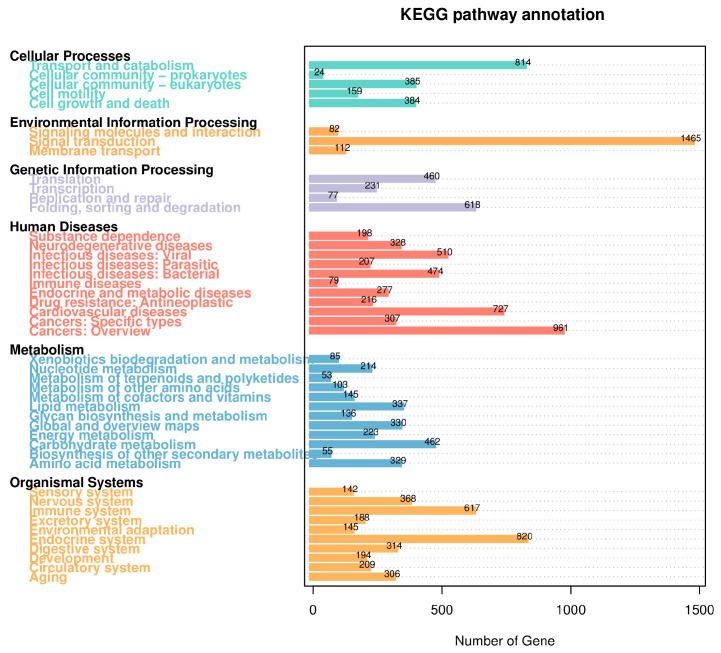
KEGG pathway classification of *B. horsfieldi* transcripts. The x-axis represents the number of transcripts, and the y-axis represents the KEGG pathway category.

**Figure 7 insects-14-00625-f007:**
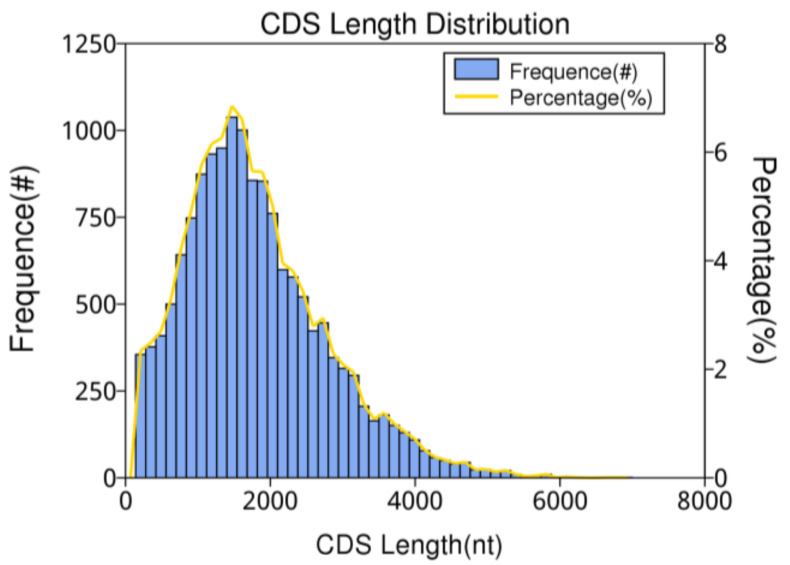
The number, percentage, and length distribution of coding sequences of *B. horsfieldi* transcripts.

**Figure 8 insects-14-00625-f008:**
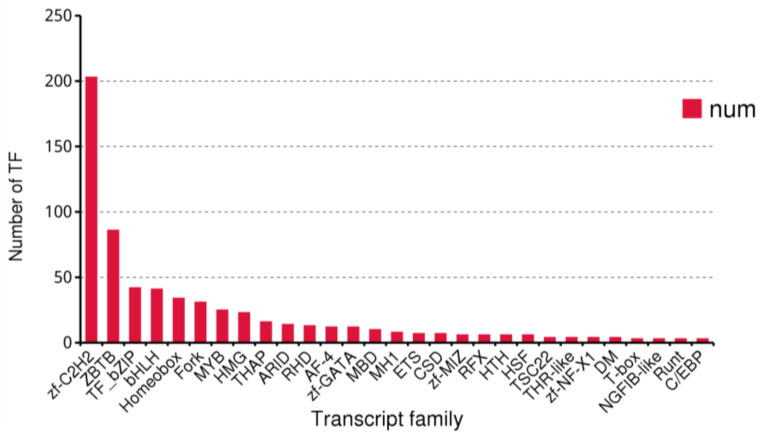
The number and family of the top 29 predicted transcription factors by SMRT.

**Figure 9 insects-14-00625-f009:**
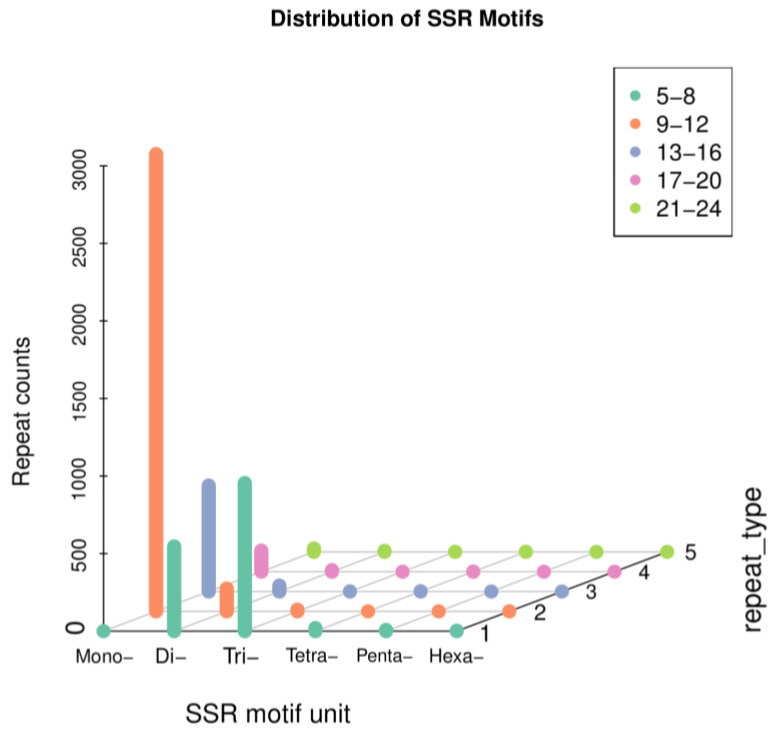
Scatter plot of simple sequence repeats in *B. horsfieldi* transcripts.

**Figure 10 insects-14-00625-f010:**
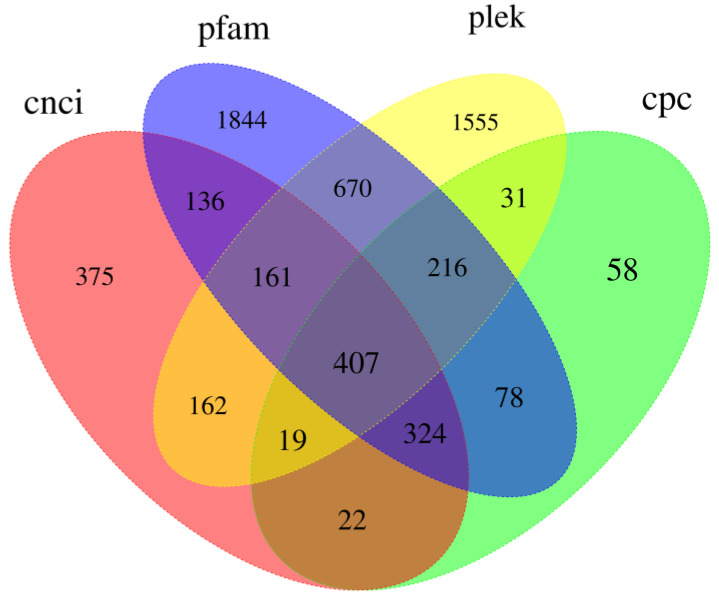
Venn diagram of identified lncRNA transcripts in *B. horsfieldi* using PLEK, CNCI, CPC, and Pfam.

**Figure 11 insects-14-00625-f011:**
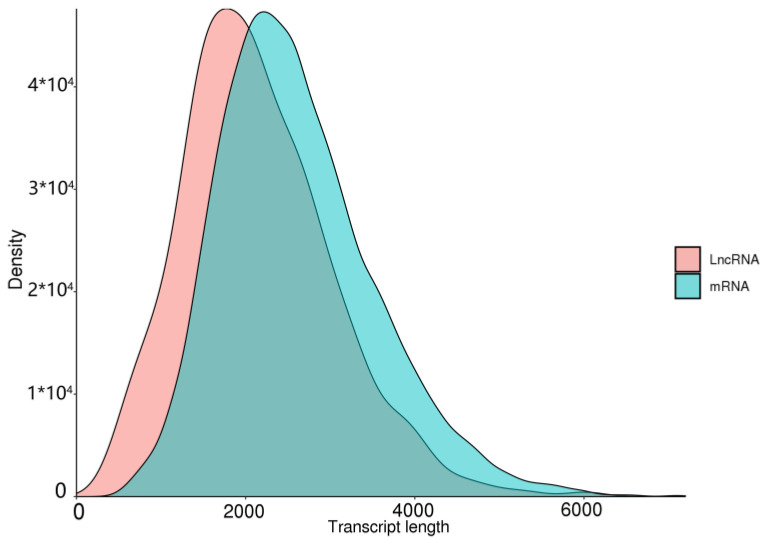
Length distribution of lncRNAs and mRNAs in *B. horsfieldi*.

**Table 1 insects-14-00625-t001:** Statistics of sequencing data and transcript clustering data.

Data Type	Total Bases (bp)	Total Number	Mean Length	Min_Length	Max_Length	N50
polymerase read	47.83 G	462,134	103,503	-	-	165,195
subread	46.31 G	20,356,793	2275	-	-	2465
CCS	-	432,091	2527	62	18,215	2674
FLNC	-	395,851	2407	86	14,739	2555

**Table 2 insects-14-00625-t002:** Quality information for sequencing data output for the samples.

Sample	Library Name	Raw Reads	Clean Reads	Raw Base (G)	Clean Base (G)	Effective (%)	Error (%)	Q20 (%)	Q30 (%)	GC (%)
BHM	FRAS220000762-4r	48699628	45453468	7.3	6.82	93.33	0.03	97.65	93.02	36.65
BHM1	FRAS220000763-4r	56680480	52076758	8.5	7.81	91.88	0.03	97.74	93.17	36.37
BHM2	FRAS220000764-3r	39332558	36671048	5.9	5.5	93.23	0.03	97.46	92.46	32.88
BHF	FRAS220000759-4r	43125796	40408496	6.47	6.06	93.7	0.03	97.87	93.26	34.54
BHF1	FRAS220000760-5r	42298740	39126102	6.34	5.87	92.5	0.03	97.77	93.14	35.24
BHF2	FRAS220000761-4r	44239794	41348904	6.64	6.2	93.47	0.03	97.76	93.13	34.36

BHM stands for *B. horsfieldi* male, and BHF stands for *B. horsfieldi* female.

**Table 3 insects-14-00625-t003:** Comparison of data before and after redundancy reduction in transcripts.

Transcript Length Interval	<500 bp	500–1 kbp	1 k–2 kbp	2 k–3 kbp	>3 kbp	Total
Number of transcripts	21	800	13,054	16,657	9380	39,912
Number of genes	3	230	3930	6593	4477	15,233

**Table 4 insects-14-00625-t004:** The number of genes corresponding to the transcripts. 1/2/3/4/5/6/7/8/9/10: number of genes containing the same number of transcripts.

**Isoform** **number**	1	2	3	4	5	6	7	8	9	10
**Unigene** **number**	10,226	2059	887	523	313	223	159	137	86	620

## Data Availability

The data supporting the results are available in a public repository at https://doi.org/10.6084/m9.figshare.22689796.v1, accessed on 25 April 2023.

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
