# Peer review of "SMRT Sequencing Technology Was Used to Construct the Batocera horsfieldi (Hope) Transcriptome and Reveal Its Features"

_insects, 2023, doi:10.3390/insects14070625_

Round 1
Reviewer 1 Report
Summary: The authors wrote a very detailed manuscript on how SMRT technology could be used to analyze transcriptome data of the B. horsfieldi pest and used it construct a type of reference genome that could be used to further study this insect pest and potentially be utilized in other eukaryotic studies. Overall, the paper is in good quality, but I do have some questions and some input:
Line 121: Under what rearing conditions were the samples taken from, please note this.
Line 121: was there a reason your mix male and female extracted RNA together? Would it be possible to do separate male and female studies?
Line 153: here is a reference for the quiver algo you could input: Nonhybrid, finished microbial genome assemblies from long-read SMRT sequencing data. (Chin et al., 2013) If you need it.
Question: Was your bioinformatics and predictive workflow done on a personal computer or a high-performance computing cluster? I’m just wondering if someone would want to do this on their own PC or would need access to an HPC.
Was this al done in one workflow or was it several workflows?
Line 179: Should this be genes?
Question: If its possible, could you use ggplot or your code to make the text bolder in the figures just so its easier to read?
Line 179: Should MISR be MISA?
Overall, I have no strong comments on the quality of english language in this. Just read through the text again and check for typos and grammatical errors.
Author Response
Dear Reviewer:
Thank you for your comments concerning our manuscript entitled " SMRT sequencing technology was used to construct the Batocera horsfieldi (Hope) transcriptome and reveal its features"(Insects-2439487). Those comments are all valuable and very helpful for revising and improving our paper, as well as the important guiding significance to our research. We have studied the comments carefully and have made corrections which we hope to meet with approval. The main corrections in the paper and the response to your comments are as flowing:
For Reviewer 1
Comments and Suggestions for Authors,
- "Under what rearing conditions were the samples taken from, please note this.” (Line 121)
Author's response: Thanks for your valuable comment. The rearing conditions of the samples have already been included in the original text. (Line 123-128, in the revised manuscript)
- "was there a reason your mix male and female extracted RNA together? Would it be possible to do separate male and female studies?” (Line 121)
Author's response: Thanks for your meaningful comment. We mix female and male individuals to extract RNA, as pooling the mixed samples of female and male adults can avoid gender interference. (Line 129-130, in the revised manuscript)
- "here is a reference for the quiver algo you could input: Nonhybrid, finished microbial genome assemblies from long-read SMRT sequencing data. (Chin et al., 2013) If you need it.” (Line 153)
Author's response: Thanks for your meaningful comment. References have been added for the Quiver algo. (Line 184 in the revised manuscript)
- "Question: Was your bioinformatics and predictive workflow done on a personal computer or a high-performance computing cluster? I’m just wondering if someone would want to do this on their own PC or would need access to an HPC.”
Was this al done in one workflow or was it several workflows?
Author's response: Thanks for your meaningful comment. It’s working on a high-performance computing cluster with several workflows.
- "Should this be genes?” (Line 179)
Author's response: Thanks for your meaningful comment. The term "genes" has been removed, and the sentence has been revised accordingly. (Line 224-225, in the revised manuscript)
- "Question: If its possible, could you use ggplot or your code to make the text bolder in the figures just so its easier to read?”
Author's response: Thanks for your meaningful comment. The text in the image has been bolded and enlarged.
- "Should MISR be MISA?” (Line 179)
Author's response: Thanks for your meaningful comment. "MISR" has been corrected to "MISA". (Line 224, in the revised manuscript)
We tried our best to improve the manuscript and made some changes in the manuscript. These changes will not influence the content and framework of the paper.
We appreciate for your warm work earnestly and hope that the correction will meet with approval.
Once again, thank you very much for your comments and suggestions.
Regards,
Xinju Wei
College of Life Science, China West Normal University,
1 Shida Road, Nanchong, 637002, China
[Email] weixinjuxx@foxmail.com

Reviewer 2 Report
This manuscript characterizes transcripts of B.horsfieldi and reveals the responses of B.horsfieldi to host plants at the transcription level by using single-molecule real-time sequencing (SMRT) and Illumina RNA-seq technologies. The study aimed to improve the genome annotation information of B.horsfieldi and facilitate the understanding of gene regulation mechanisms. The scope of this research fits the scope of this journal. However, this work does not clearly clarify the biological significance of transcriptome features in the interactions between B.horsfieldi and its host plants. To meet a standard requirement for insects, the manuscript also needs to obtain more detailed mechanisms to understand the transcriptome features of B.horsfieldi. There are several concerns that need to be clarified as follows.
MAJOR COMMENTS
1. It was mentioned in the introduction and discussion that SMRT technology is more suitable for alternative splicing events and fusion gene analyses, but no corresponding analysis was done in the results.
2. The title is a bit inappropriate. In this study, the transcriptome features of B. horsfieldi were analyzed using RNA-seq and Iso-seq. It would be clearer if " reference genome " were removed from the title.
3. In the introduction part, it would be better to provide some more information on the transcriptome features of B.horsfieldi response to different host species. Whether these changes in transcription levels are included in the transcriptome data studied in this paper, this information can be described in the discussion.
4. In the method part, some detailed information could be provided. For instance, in ‘Sample Collection and Preparation’, how many male and female insects were used for RNA extraction? How many biological repeats have been used in Iso-seq and RNA-seq? Whether the samples collected in this study are representative of the B. horsfieldi species.
5. Again, in the method part, B. horsfieldi has a wide variety of host species, in this study, the insects were collected from which host plant? Were the insects collected in the wild or raised under laboratory conditions? What are the conditions?
6. It is unclear how reads were mapped. Which program and settings were used? Which reference genome was used? The authors should provide more details.
7. It would be better to provide a figure to brief summary of how the individuals for Iso-seq and RNA-seq were prepared.
8. In GO, KEGG, KOG, and TF annotations, only the number of annotated genes and the top terms were displayed, suggesting a further interpretation of these genes and terms with higher abundance.
9. In each section, after describing the sequencing data, try to draw a biological conclusion.
MINOR COMMENTS
1. Line 31/line 208/line 204/line 226: FLNC, ZMW, BHM, BHF, the full name of an abbreviation should be provided when it first appears.
2. Line 38: change "RNAi" to "gene editing". RNAi is one of the methods of gene editing, but "RNAi" is not comprehensive enough.
3. Line 143/line 168/line 169/line 190/line 212/line 217: The software version used in this study should be complete.
4. Line 232, Fig. 1: The bar chart is difficult to show the intersection of annotated genes in each database, and it is better to replace it with the Wayne chart to more fully reflect the number of unique and common annotated genes in each database.
5. Line 215: In ‘Sample Collection and Preparation’, the mixed whole sample was used for subsequent RNA sequencing. What does the "two samples" mean here?
Author Response
Dear Reviewer:
Thank you for your comments concerning our manuscript entitled " SMRT sequencing technology was used to construct the Batocera horsfieldi (Hope) transcriptome and reveal its features"(Insects-2439487). Those comments are all valuable and very helpful for revising and improving our paper, as well as the important guiding significance to our research. We have studied the comments carefully and have made corrections which we hope to meet with approval. The main corrections in the paper and the response to your comments are as flowing:
For Reviewer 2
Specific comments,
- "It was mentioned in the introduction and discussion that SMRT technology is more suitable for alternative splicing events and fusion gene analyses, but no corresponding analysis was done in the results.”
Author's response: Thanks for your valuable comment. This portion of the results is currently undergoing further in-depth research and will be used for publication in another paper.
- "The title is a bit inappropriate. In this study, the transcriptome features of horsfieldi were analyzed using RNA-seq and Iso-seq. It would be clearer if " reference genome " were removed from the title.”
Author's response: Thanks for your valuable comment. The phrase "reference genome" has been removed, and the title has been appropriately revised.
- "In the introduction part, it would be better to provide some more information on the transcriptome features of horsfieldi response to different host species. Whether these changes in transcription levels are included in the transcriptome data studied in this paper, this information can be described in the discussion.”
Author's response: Thanks for your valuable advice. The characteristics of B. horsfieldi's response to different host plant species have been described in the Discussion section. (Line 501-504, in the revised manuscript)
- "In the method part, some detailed information could be provided. For instance, in ‘Sample Collection and Preparation’, how many male and female insects were used for RNA extraction? How many biological repeats have been used in Iso-seq and RNA-seq? Whether the samples collected in this study are representative of the horsfieldi species.”
Author's response: Thanks for your valuable advice. Revisions have been made to the relevant issues in the "Sample Collection and Preparation" section. The samples collected in this study are representative of the species B. horsfieldi. (Line 123-128, in the revised manuscript)
- "Again, in the method part, horsfieldi has a wide variety of host species, in this study, the insects were collected from which host plant? Were the insects collected in the wild or raised under laboratory conditions? What are the conditions?”
Author's response: Thanks for your valuable advice. The insects used in this study were collected from the host plant Populus tomentosa, which is the host of B. horsfieldi. The rearing conditions for the insects are described in lines 123-128 of the revised manuscript.
- "It is unclear how reads were mapped. Which program and settings were used? Which reference genome was used? The authors should provide more details.”
Author's response: Thanks for your valuable advice. Since it is a de novo transcriptome analysis, there is no need for a reference genome. For the quantification step, the reference used is the de-redundant transcript sequence. The software used for read alignment and quantification is as follows:
bowtie2 V2.3.4 |
-qï¼›--phred33ï¼›--sensitiveï¼›--dpad 0ï¼›--gbar 99999999ï¼›--mp 1,1ï¼›--np 1ï¼›--score-min L,0,-0.1ï¼›-I 1ï¼›-X 1000ï¼›--no-mixedï¼›--no-discordantï¼›-p 8ï¼›-k 30 |
Transcript alignment. |
http://bowtie-bio.sourceforge.net/bowtie2/index.shtml |
RSEM V1.3.0 |
--phred33ï¼›-qualsï¼›--forward-prob 0.5ï¼›--time |
Transcript quantification analysis |
http://deweylab.github.io/RSEM/ |
- "It would be better to provide a figure to brief summary of how the individuals for Iso-seq and RNA-seq were prepared.”
Author's response: Thanks for your valuable advice. The workflow diagrams for Iso-seq and RNA-seq are shown below.
|
|
- "In GO, KEGG, KOG, and TF annotations, only the number of annotated genes and the top terms were displayed, suggesting a further interpretation of these genes and terms with higher abundance.”
Author's response: Thanks for your valuable advice. We utilized four databases, namely GO, KEGG, KOG, and TF, to effectively demonstrate the enrichment of the data. By employing these databases, we can express gene expression levels more precisely and clearly.
- "In each section, after describing the sequencing data, try to draw a biological conclusion.”
Author's response: Thanks for your valuable advice. The biological conclusions have been added.
Comments on the Quality of English Language,
MINOR COMMENTS,
- Line 31/line 208/line 204/line 226: FLNC, ZMW, BHM, BHF, the full name of an abbreviation should be provided when it first appears.
Author's response: Thanks for your valuable advice. "FLNC, ZMW, BHM, and BHF” have been revised. (Lines 31, 262, and 286, in the revised manuscript)
- Line 38: change "RNAi" to "gene editing". RNAi is one of the methods of gene editing, but "RNAi" is not comprehensive enough.
Author's response: Thanks for your valuable advice. "RNAi" has been changed to "gene editing". (Line 38 in the revised manuscript)
- Line 143/line 168/line 169/line 190/line 212/line 217: The software version used in this study should be complete.
Author's response: Thanks for your valuable advice. The relevant software versions have been added.
- Line 232, Fig. 1: The bar chart is difficult to show the intersection of annotated genes in each database, and it is better to replace it with the Wayne chart to more fully reflect the number of unique and common annotated genes in each database.
Author's response: Thanks for your valuable advice. We used bar graphs to present the length and quantity of unigenes more clearly, while Venn diagrams were not able to represent the length of unigenes.
- Line 215: In ‘Sample Collection and Preparation’, the mixed whole sample was used for subsequent RNA sequencing. What does the "two samples" mean here?
Author's response: Thanks for your valuable advice. These two mixed samples appear in the "Materials and Methods" section and are mixed RNA samples of male and female adults of B. horsfieldi. (Lines 128-132 in the revised manuscript)
We tried our best to improve the manuscript and made some changes in the manuscript. These changes will not influence the content and framework of the paper.
We appreciate for Editors' and Reviewers' warm work earnestly and hope that the correction will meet with approval.
Once again, thank you very much for your comments and suggestions.
Regards,
Xinju Wei
College of Life Science, China West Normal University,
1 Shida Road, Nanchong, 637002, China
[Email] weixinjuxx@foxmail.com
